# A comparison of self-triage tools to nurse driven triage in the emergency department

**Sachin V. Trivedi** [ID] [1] *, **Rachit Batta** [2], **Nicolas Henao–Romero** [3], **Prosanta Mondal** [4], **Tracy Wilson** [1], **James Stempien** [1]

1 Department of Emergency Medicine, University of Saskatchewan, Saskatoon, Saskatchewan, Canada,
2 Department of Emergency Medicine, University of Calgary, Calgary, Alberta, Canada, 3 College of Medicine, University of Saskatchewan, Saskatoon, Saskatchewan, Canada, 4 Department of Community Health and Epidemiology, University of Saskatchewan, Saskatoon, Saskatchewan Canada

* Sachin.trivedi@usask.ca

## Abstract

### Introduction

Canadian patients presenting to the emergency department (ED) typically undergo a triage process where they are assessed by a specially trained nurse and assigned a Canadian Triage and Acuity Scale (CTAS) score, indicating their level of acuity and urgency of assessment. We sought to assess the ability of patients to self-triage themselves through use of one of two of our proprietary self-triage tools, and how this would compare with the standard nurse-driven triage process.

### Methods

We enrolled a convenience sample of ambulatory ED patients aged 17 years or older who presented with chief complaints of chest pain, abdominal pain, breathing problems, or musculoskeletal pain. Participants completed one, or both, of an algorithm generated self-triage (AGST) survey, or visual acuity scale (VAS) based self-triage tool which subsequently generated a CTAS score. Our primary outcome was to assess the accuracy of these tools to the CTAS score generated through the nurse-driven triage process.

### Results

A total of 223 patients were included in our analysis. Of these, 32 (14.3%) presented with chest pain, 25 (11.2%) with shortness of breath, 75 (33.6%) with abdominal pain, and 91 (40.8%) with musculoskeletal pain. Of the total number of patients, 142 (47.2%) completed the AGST tool, 159 (52.8%) completed the VAS tool and 78 (25.9%) completed both tools. When compared to the nurse-driven triage standard, both the AGST and VAS tools had poor levels of agreement for each of the four presenting complaints.

### Conclusions

Self-triage through use of an AGST or VAS tool is inaccurate compared to the established standard of nurse-driven triage. Although existing literature exists which suggests that self-triage tools developed for specific subsets of complaints may be feasible, our results would

**Data Availability Statement:** De-identified data is available here: https://dataverse.harvard.edu/dataset.xhtml?persistentId=doi:10.7910/DVN/MGDOWQ.

**Funding:** This study received research grant funding of $21, 315 in 2018 from the Royal University Hospital Foundation in Saskatoon, Saskatchewan. Of those authors awarded this funding, ST and JS remained with the project and contributed the requisite amount for authorship. None of the authors received any salary from this. A link to the RUH Foundation can be found here: https://ruhf.org. The funders had no role in study design, data collection and analysis, decision to publish, or preparation of the manuscript.

**Competing interests:** The authors have declared that no competing interests exist

suggest that adopting the self-triage approach on a broader scale for all-comers to the ED does not appear to be a viable option to enhance the current triage process. Further study is required to show if self-triage can be used in the ED to optimize the triage process.

## Introduction

When patients present to the emergency department (ED), triage serves as their gateway to receiving care. The current model of triage involves the patient being assessed by a specially trained registered nurse wherein they are asked a series of questions relating to their chief complaint. This process results in a score which assigns a level of acuity and urgency to a given patient's presenting complaint. The Canadian Triage and Acuity Scale (CTAS) is the reference standard for this output in Canada. The current landscape of emergency medicine is mired by prolonged wait times, which are known to have negative patient outcomes [1, 2]. As it currently stands, the current process of patient triage is done without the patient providing any information prior to the nursing assessment. In time of high ED volumes and patient registrations, incorporating the patient into the triage process may serve as a patient-centered innovation to enhance its efficiency by having the patient provide their subjective part of the triage assessment in advance of the triage nurse's objective assessment.

This notion of self-triage has been studied previously with success in specific populations [3–5]. However, this approach may be fraught with challenges owing to individual health literacy. Previous literature would suggest that laypersons may be overcautious in deciding their need to seek medical care, or miss identifying potential emergencies [6]. Indeed, laypersons have also been previously demonstrated to not follow prescribed advice when directed to either present, or not present, to an ED to seek care [6–8]. We previously sought to compare a computer-assisted self-triage tool to the traditional method of triage and found low rates of agreement and high rates of over triage, a result consistent with other literature [4, 6, 9]. The inconsistency of results in the existing knowledge base indicates that though self-triage has potential to enhance the overall triage process, it is not quite ready for a more widespread adoption to all-comers to the ED.

In this study, we aimed to build on the existing knowledge base and further our previous work by assessing the ability for patients to self-triage. In our previous work, we had limited involvement of patient and family partners as well as registered nurses in the development of our computer-assisted self-triage tool [9]. As such, we made meeting with these key stakeholders a priority in this study and reviewed our methods of self-triage with them on an iterative basis. These were an algorithm-generated self-triage (AGST) score and a visual analogue scale based (VAS) score. We based this on four specific presenting complaints: chest pain, shortness of breath, abdominal pain and musculoskeletal pain. In an attempt to see if this is a viable option for the evolution of triage, we aimed to compare the accuracy of these tools against the gold standard of nurse driven triage with respect to their resulting CTAS score.

## Methods

### Setting

This was a prospective, observational pilot study conducted within the tertiary care EDs at Royal University Hospital and St. Paul's Hospital in Saskatoon, Saskatchewan, Canada. We enrolled a convenience sample of ambulatory patients who presented either independently or

with a caregiver that were aged 17 and older. We selected for those who presented with chest pain, abdominal pain, shortness of breath, and musculoskeletal chief complaints. Those excluded were the patients who presented by ambulance and who had different presenting complaints as listed above. We further excluded those patients who identified discrepant presenting complaints between our self-triage tools and the nurse driven triage, as well as those who left the ED prior to completing the formal triage process.

Eligible participants were identified by a research assistant who was present in the ED every afternoon, or early evening, for five hours at a time from Monday through Friday beginning on November 4, 2019. Enrollment of participants was stopped due to the beginning of the Covid-19 pandemic in March 11, 2020. Once identified, the research assistant explained the purpose of our project. Participants subsequently had the option to opt in or out of the project. This project was funded by the Royal University Hospital Foundation. The University of Saskatchewan's Research Ethics Board granted approval for this project (BEH #16–239). Operational approval for this study was obtained from the Saskatchewan Health Authority.

## Intervention and outcome

We designed two tools which were used to generate self-triage CTAS scores. An emergency physician and two emergency medicine residents drafted the initial tools, which were then vetted by a heterogenous group of individuals in an iterative process. This group included a departmental research facilitator, senior ED nurses, and representatives of the Saskatchewan Health Authority's Patient and Family Advisory Council. Ultimately, the two tools took the form of an AGST survey, and a VAS based tool. Both tools provided a CTAS score output, as described below. For the purposes of this project, we combined the higher acuity CTAS 1 and CTAS 2 patients in the same grouping. Similarly, we grouped CTAS 4 and 5 patients together.

The AGST survey (seen in Table 1) consisted of five to six symptoms based yes or no questions depending on presenting complaint. If patients answered "yes" to one or two questions, patients were assigned as being either a CTAS 4 or 5 presentation. Similarly, if patients answered "yes" to three or four categories, they were assigned as being a CTAS 3 presentation. Finally, if all questions were responded to as "yes," patients were assigned as being a CTAS 2.

The VAS based tool consisted of a sliding scale from 0 to 10. It asked four questions which were consistent across all presenting complaints. These were: 1) how much pain are you in right now; 2) how worried are you about your condition right now; 3) how urgently for you think you need to see the doctor; and 4) how much is your condition interfering with your daily activities. The scores for all of these were subsequently averaged. If the average VAS score was between 1 and 3.3, the patient was listed as a CTAS 4 or 5 presentation. If the average score was between 3.4 and 6.6, the patient was assigned as a CTAS 3. Finally, if the score was between 6.7 and 10, the patient was noted as a CTAS 2.

Both tools were built on the Ethica app software in order to maintain the necessary level of security and privacy required by our local research ethics board. They were loaded on four Samsung Galaxy 6 smartphones. Depending on individual comfort level, participants were given the option to handle the smartphones themselves in filling out the app, or having the research assistant input their answers for them. Three different versions of the survey were administered at random: 1) one version was a visual analog/ Likert scale; 2) the second version a series of dichotomous closed-ended questions wherein the answers were simply yes or no; and 3) the third version a combination of both scale and questions.

Participants enrolling in our study indicated their informed consent through completion of a form which was presented to them on the smartphone they were using prior to using any of our self-triage tools. Following this, they provided demographic details as well as their name

**Table 1. AGST survey questions.**

| Study Category | Questions |
|---|---|
| Chest Concerns | Does your chest [pain/discomfort/heaviness] affect your daily activities? |
| | Have you been admitted into hospital (stayed overnight in hospital) for a similar problem in the past? |
| | Does this problem cause you [shortness of breath/difficulty breathing]? |
| | Was there any trauma/injury to the area? |
| | Do you have any of these risk factors? (high cholesterol high blood pressure diabetes cigarette smoking family members with heart problems) |
| Breathing Problems | Do you have any cough/congestion? |
| | Have you had any fever or cold sweats in the last 24 hours? |
| | Does the breathing problem affect your daily activities? |
| | Does the breathing problem cause you any [chest pain/discomfort/heaviness]? |
| | Any previous medical diagnoses that have caused you breathing difficulties? (heart failure, asthma, COPD, cancer etc.) |
| | Has anything happened recently to make these symptoms worse (falls, colds, run out of medication)? |
| Abdominal Concerns | Does this problem cause you nausea/vomiting? |
| | Does your pain affect the activities you normally perform each day? |
| | Have you stayed overnight in hospital for this problem previously? |
| | Has this pain ever happened to you previously? |
| | Have you noticed any abnormal bleeding or bleeding that would otherwise not be normally present? |
| Bone and Joint Concerns | Are you still able to move your affected joint as you were able to a week ago? |
| | Have you stayed overnight in hospital for a similar problem? |
| | Can you move the area with minimal pain? |
| | Was there any injury that came before the pain? |
| | Are you able to walk? |

and birthdate. At the conclusion of the data collection period, we used the latter information to retrospectively review their chart to acquire the CTAS scores which were generated through nurse-driven triage. This retrospective review was completed on March 2, 2021.

Our primary outcome of this study was to assess the level of concordance between both sets of self-triage scores and the reference standard of the triage score assigned by the usual process of nurse driven triage.

## Sample size calculation and statistical analysis

As a pilot correlational study, we intended to have a minimum of 16 participants per each individual variable, defined here as the unique presenting complaints [10]. We sought to meet this minimum for each triage tool. Once all data was collected, it was analyzed quantitatively, using Kappa statistics calculated through the SPSS program. Kappa statistics were used to determine level of correlation between the self-triage tools and the nurse driven triage standard.

## Results

A total of 279 participants were enrolled in the study and completed the self-triage questionnaire. Of this group, 39 participants were excluded as their self-reported chief complaint did not match their actual chief complaint. An additional 17 participants were excluded as they had left prior to the formal triage process, and so no actual CTAS score could be determined. Ultimately, 223 participants were included for analysis. The emergency department of Royal

**Table 2. Demographics.**

|  |  | N (%) |
|---|---|---|
| **Sex** | **Male** | 103 (46.2) |
|  | **Female** | 120 (53.8) |
| **Age (Years)** | **<20** | 15 (6.7) |
|  | **20–39** | 120 (53.8) |
|  | **40–60** | 57 (25.6) |
|  | **>60** | 31 (13.9) |
| **Presenting Complaint** | **Chest Concerns** | 32 (14.3) |
|  | **Breathing Concerns** | 25 (11.2) |
|  | **Abdominal Concerns** | 75 (33.6) |
|  | **Bone and Joint Concerns** | 91 (40.8) |
| **Completed Self-Triage Tool** | **Algorithm-generated self-triage (AGST)** | 142 (47.2) |
|  | **Visual Acuity Scale (VAS)** | 159 (52.8) |
|  | **Both AGST and VAS** | 78 (25.9) |

University Hospital provided the majority of the data, as the emergency department of Saint Paul's Hospital was often empty during the months in which it was attended or devoid of patients whose conditions were appropriate for the study.

With respect to the different forms of our self-triage tools, a total of 159 participants completed the VAS based tool, and 142 participants completed the AGST survey. Of these participants, 78 completed both self-triage tools. Of the four study categories, chest concerns comprised 14.3% (n = 32) of the responses, breathing concerns comprised 11.2% (n = 25) of the responses, abdominal concerns comprised 33.4% (n = 75) of the responses and musculoskeletal concerns comprised 40.8% (n = 91) of the responses.

Table 2 outlines the descriptive statistics of our study. Table 3 displays the concordance of the AGST tool with the nurse-driven triage result, and Table 4 displays the same but with the VAS tool. Table 5 presents the data of the subgroup which completed both the AGST and VAS tools.

## Discussion

In this study, we endeavored to see if ambulatory patients who presented to the ED could use our proprietary self-triage tools to similar acuity levels in comparison to the reference standard of nurse-driven triage. Our intent with this study was to see if the triage process could evolve in a patient-centered format. Our results demonstrate that, in their current form, both the AGST survey and VAS tools are not appropriate for this purpose. In the case of participants who individually completed either tool, agreement was non-existent, minimal, or weak at best. Though the existing literature has demonstrated a high degree of over-triage [4, 6, 9], in our study patients both over- and under-triaged themselves. The AGST survey tool appeared to

**Table 3. AGST tool vs. nurse-driven triage.**

| Presenting Complaint | Total (%) | Agreement (%) | Over-triage (%) | Under-triage (%) | Kappa Value |
|---|---|---|---|---|---|
| **Chest Concerns** | 23(16.2) | 11(47.8) | 10(43.5) | 2(8.7) | 0.156 |
| **Breathing Problems** | 16(11.3) | 10(62.5) | 2(12.5) | 4(25) | 0.273 |
| **Abdominal Concerns** | 43(30.3) | 21(48.8) | 13(30.2) | 9(20.9) | -0.019 |
| **Bone and Joint Concerns** | 60(42.3) | 41(68.3) | 6(10.0) | 13(21.7) | 0.116 |

**Table 4. VAS triage tool vs. nurse-driven triage.**

| Presenting Complaint | Total (%) | Agreement (%) | Over-triage (%) | Under-triage (%) | Kappa Value |
|---|---|---|---|---|---|
| Chest Concerns | 22(13.8) | 7(31.8) | 8(36.4) | 7(31.8) | -0.111 |
| Breathing Problems | 20(12.6) | 9(45) | 2(10) | 9(45) | 0.083 |
| Abdominal Concerns | 56(35.2) | 19(33.9) | 5(8.9) | 32(57.1) | 0.015 |
| Bone and Joint Concerns | 61(38.4) | 9(14.8) | 1(1.6) | 51(83.6) | -0.028 |

slightly outperform the VAS scale, including in the group of participants who completed both self-triage tools. Though it is important to understand the subjective experience of the patient with respect to treating their symptoms and managing their expectations in a given ED visit, it is likely that the subjectivity of the responses from the VAS score influenced the VAS tool's relative underperformance.

We had hoped that by this degree of patient involvement in triage, the triage component in ED-experience for patients could be optimized. Self-assessments have previously demonstrated some success with triage when they are honed down to specific complaints [3, 5, 11]. One particular ED based study looked at patients presenting with chest pain used a similar "yes" or "no" based approach like our AGST tool and had high levels of agreement with low risk to patient safety [11]. The success of these studies raises the question of how focused, complaint–specific tools may be utilized to optimize triage overall. Kiosk based triage, though largely focused on early registration of patients, has been studied in terms of optimizing efficiency in triage [12, 13]. In considering this, it follows that combining kiosk based triage with the development of individual, complaint-focused tools may represent an area for how triage can evolve.

Innovation in triage is not limited to this, however, as other technologies have potential for use. Previously, online-symptom checkers have been studied with respect to their overall diagnostic and triage accuracy, but these have similarly been faced with challenges surrounding accuracy, and rates of over or under triage [14–16]. Artificial intelligence may also have uses in triage, with machine learning models having been demonstrated as having some success with specific presenting complaints [17–19]. The possibilities of how triage can evolve can take a variety of different forms, and these merit further investigation.

Although our study had a negative result, we are confident that it provides additional context to studying the future of ED triage. Our study design was strengthened by the fact that we tested two different self-triage tools and included a group where patients completed both tools, comparing their accuracy against each other with a controlled userbase. Further, we had involved patient and family advisors in the development process so as to capture their insights. We do note that approximately 6% of our patients had left the ED prior to making contact

**Table 5. Subgroup analysis of participants who completed both AGST and VAS tools vs. nurse-driven triage.**

| Presenting complaint (n, %) | Self–Triage tool used | Agreement (%) | Over-triage (%) | Under-triage (%) | Kappa Statistic (95% CI) |
|---|---|---|---|---|---|
| Chest Concerns (n = 13, 16.7) | AGST | 7(53.8) | 5(38.5) | 1(7.7) | 0.264 |
| | VAS | 7(53.8) | 4(30.8) | 2(15.4) | 0.235 |
| Breathing Problems (n = 11, 14.1) | AGST | 7(63.6) | 2(18.2) | 2(18.2) | 0.302 |
| | VAS | 6(54.5) | 0(0) | 5(45.5) | 0.295 |
| Abdominal Concerns (n = 24, 30.8) | AGST | 14(58.3) | 6(25) | 4(16.7) | 0.172 |
| | VAS | 9(37.5) | 0(0) | 15(62.5) | 0.100 |
| Bone and Joint Concerns (n = 30, 38.5) | AGST | 20(66.7) | 4(13.3) | 6(20) | 0.091 |
| | VAS | 4 (13.3) | 0 (0) | 26 (86.7) | -0.04 |

with the triage nurse. We do not believe that our innovations contributed to this as this appears to be in line with our unpublished institutional rates of patients leaving without being seen. Furthermore, as our study was taking place during the beginning of the Covid-19 pandemic, one could speculate that this also contributed to patients leaving the ED without being triaged.

We do recognize that our study had some limitations. Recruitment of our participants was done by way of convenience sampling and this likely introduced some bias as our data collection was predominantly done throughout daytime and weekday hours. Furthermore, bias from this recruitment method was also introduced by the majority of our patients being from the Royal University Hospital, meaning that our sample was likely not balanced with those socially disadvantaged patients who typically present to our inner-city hospital. We feel that convenience sampling likely led to a study population which was of a higher level of affluence or health literacy. As well, although we had stated that participation in our study would not result in any expedited assessments in the ED, patients may have still intentionally over-triaged themselves in order to avoid waiting for a prolonged period. Finally, though nurse-driven triage was our reference standard, this is an imperfect system with established variability [20, 21].

## Conclusions

Ultimately, our study showed that self-triage through use of an AGST or VAS tool does not appear to be a viable option for use in the ED. Further studies looking to assess feasibility of self-triage tools could look at utilizing focused questionnaires designed around specific presenting complaints. Leveraging existing technology, such as kiosk -based triage tools, may also represent an area for future studies on how to optimize triage. Finally, the burgeoning use of artificial intelligence in health care may provide an opportunity to overhaul triage for ED patients.

## Supporting information

**S1 Data. Deidentified data.** This file contains our deidentified data set.
(XLSX)

## Acknowledgments

The authors would like to thank Thomas Graham, Marissa Evans, Jenna Mee and Nathaniel Osgood for their contributions to this study.

## Author Contributions

**Conceptualization:** Sachin V. Trivedi, Rachit Batta, James Stempien.

**Data curation:** Sachin V. Trivedi, Rachit Batta, Nicolas Henao–Romero, Tracy Wilson.

**Formal analysis:** Sachin V. Trivedi, Rachit Batta, Nicolas Henao–Romero, Prosanta Mondal, Tracy Wilson, James Stempien.

**Funding acquisition:** Sachin V. Trivedi, James Stempien.

**Investigation:** Sachin V. Trivedi, Rachit Batta, James Stempien.

**Methodology:** Sachin V. Trivedi, Rachit Batta, James Stempien.

**Project administration:** Sachin V. Trivedi, James Stempien.

**Supervision:** James Stempien.

**Writing – original draft:** Sachin V. Trivedi, Rachit Batta, Nicolas Henao–Romero, Prosanta Mondal, Tracy Wilson, James Stempien.

**Writing – review & editing:** Sachin V. Trivedi, Rachit Batta, Nicolas Henao–Romero, Prosanta Mondal, Tracy Wilson, James Stempien.

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
