## [Decision Letter · Decision Letter 0]

7 May 2024

PONE-D-23-43949A comparison of self-triage tools to nurse driven triage in the emergency departmentPLOS ONE

Dear Dr. Trivedi,

Thank you for submitting your manuscript to PLOS ONE. After careful consideration, we feel that it has merit but does not fully meet PLOS ONE’s publication criteria as it currently stands. Therefore, we invite you to submit a revised version of the manuscript that addresses the points raised during the review process.

We look forward to receiving your revised manuscript.

Kind regards,

Sebastian Schnaubelt, MD, PhD

Academic Editor

PLOS ONE

3. In the online submission form, you indicated that [As we accessed patient data to acquire their nurse-driven triage scores, we cannot release the full data set publically given how this contains identifying information. Should a reviewer want to access the data, we could provide anonymized data upon request.]. 

Reviewers' comments:

Reviewer's Responses to Questions

**Comments to the Author**

1. Is the manuscript technically sound, and do the data support the conclusions?

Reviewer #1: Yes

Reviewer #2: Yes

2. Has the statistical analysis been performed appropriately and rigorously? 

Reviewer #1: Yes

Reviewer #2: No

3. Have the authors made all data underlying the findings in their manuscript fully available?

Reviewer #1: Yes

Reviewer #2: No

4. Is the manuscript presented in an intelligible fashion and written in standard English?

Reviewer #1: Yes

Reviewer #2: Yes

5. Review Comments to the Author

Reviewer #1: Thank you for asking me to review the manuscript PONE-D-23-43949, A comparison of self-triage tools to nurse driven triage in the emergency department

The statistical analyses are appropriate. The style of the paper is appropriate according to the journal’s requirements. Appreciate the authors best efforts for testing a self-triage tool due to increasing digitalisation and the use of artificial intelligence; no further comments

Reviewer #2: Thanks for the opportunity to review this novel study

Abstract

• The abstract provides a concise overview of the study's purpose, methods, and key findings. However, it could be enhanced by briefly stating the significance of the findings in the context of existing literature or potential implications for emergency department operations.

• Include a sentence on the implications of the findings and a brief comparison with existing literature to contextualize the significance of the study.

Background

• The introduction clearly establishes the background, but I am not clear on the “need” for this study. What is the cost or downside of the current model? References are well integrated to justify the research.

• The introduction could benefit from a more detailed discussion on the limitations of previous studies and how this study aims to address them.

Methods

• The description of the methodology is detailed, including sample selection, intervention tools, and data collection processes which are crucial for replicability.

• There is a clear explanation of the statistical methods used, which is appropriate for the study design. However, would an agreement study be a better design? Who is to say what the right triage category is? It would be interesting to link the scores (AGST and CTAS) to an objective marker of ED clinical needs.

Results

• The results are presented in a clear, logical order. Tables are appropriately used to summarize data, making it easy to understand.

• The study does well to present the quantitative data; however, there is a lack of depth in discussing unexpected results or discrepancies.

• What are the reasons behind the poor agreement levels of the self-triage tools compared to CTAS? Could you run some analysis to shine light on this?

Discussion

• The discussion contextualizes the results within the broader literature and discusses potential reasons for the findings. It also acknowledges the study's limitations and suggests areas for future research.

• This is a convenience sample – a limitation that should be explored in more detail.

• More than 6 % left before they had a CTAS – this seems very high. Did the intervention cause this level of left before seen or is this the base rate at your facility?

• The discussion remains consistent with the results, and does not overstate the findings.

• Suggest providing more detailed suggestions for future research directions and potential practical applications of the study findings in real-world settings.

Conclusion

• Please separate out from the discussions

6. PLOS authors have the option to publish the peer review history of their article (what does this mean?). If published, this will include your full peer review and any attached files.

Reviewer #1: No

Reviewer #2: No

---

## [Author Response · Author response to Decision Letter 0]

8 Jul 2024

Please see attached document containing response to reviewers

---

## [Editor Report · Decision Letter 1]

12 Aug 2024

A comparison of self-triage tools to nurse driven triage in the emergency department

PONE-D-23-43949R1

Dear Dr. Trivedi,

We’re pleased to inform you that your manuscript has been judged scientifically suitable for publication and will be formally accepted for publication once it meets all outstanding technical requirements.

Kind regards,

Sebastian Schnaubelt, MD, PhD

Academic Editor

PLOS ONE
---

## [Editor Report · Acceptance letter]

16 Aug 2024

PONE-D-23-43949R1 

PLOS ONE

Dear Dr. Trivedi, 

I'm pleased to inform you that your manuscript has been deemed suitable for publication in PLOS ONE. Congratulations! Your manuscript is now being handed over to our production team.

Kind regards, 

on behalf of

Dr. Sebastian Schnaubelt 

Academic Editor

PLOS ONE